# Ex Vivo Mesenchymal Stem Cell Therapy to Regenerate Machine Perfused Organs

**DOI:** 10.3390/ijms22105233

**Published:** 2021-05-15

**Authors:** Christina Bogensperger, Julia Hofmann, Franka Messner, Thomas Resch, Andras Meszaros, Benno Cardini, Annemarie Weissenbacher, Rupert Oberhuber, Jakob Troppmair, Dietmar Öfner, Stefan Schneeberger, Theresa Hautz

**Affiliations:** Department of Visceral, Transplant and Thoracic Surgery (VTT), Center of Operative Medicine, Organ Life Laboratory and Daniel Swarovski Research Laboratory (DSL), Medical University of Innsbruck (MUI), 6020 Innsbruck, Austria; christina.bogensperger@i-med.ac.at (C.B.); julia.hofmann@i-med.ac.at (J.H.); franka.messner@i-med.ac.at (F.M.); t.resch@tirol-kliniken.at (T.R.); andras.meszaros@i-med.ac.at (A.M.); benno.cardini@i-med.ac.at (B.C.); annemarie.weissenbacher@i-med.ac.at (A.W.); rupert.oberhuber@i-med.ac.at (R.O.); jakob.troppmair@i-med.ac.at (J.T.); dietmar.oefner@i-med.ac.at (D.Ö.); stefan.schneeberger@i-med.ac.at (S.S.)

**Keywords:** regeneration, mesenchymal stem cells, machine perfusion

## Abstract

Transplantation represents the treatment of choice for many end-stage diseases but is limited by the shortage of healthy donor organs. Ex situ normothermic machine perfusion (NMP) has the potential to extend the donor pool by facilitating the use of marginal quality organs such as those from donors after cardiac death (DCD) and extended criteria donors (ECD). NMP provides a platform for organ quality assessment but also offers the opportunity to treat and eventually regenerate organs during the perfusion process prior to transplantation. Due to their anti-inflammatory, immunomodulatory and regenerative capacity, mesenchymal stem cells (MSCs) are considered as an interesting tool in this model system. Only a limited number of studies have reported on the use of MSCs during ex situ machine perfusion so far with a focus on feasibility and safety aspects. At this point, no clinical benefits have been conclusively demonstrated, and studies with controlled transplantation set-ups are urgently warranted to elucidate favorable effects of MSCs in order to improve organs during ex situ machine perfusion.

## 1. Introduction

Acute and chronic organ failure, caused by infection, cancer, or chronic disease, represents a life-threatening condition. In most cases, organ transplantation is the only effective therapy for permanent restoration of organ function. However, an increasing disparity exists between demand and supply of donor organs for transplantation, which leaves many patients suffering and eventually dying while on the waiting list [1,2,3]. Nevertheless, suboptimal organs with pre-existing damage and organs from older or cardiac death donors (DCD) are particularly susceptible to pronounced organ damage caused by ischemia/reperfusion injury (IRI), resulting in inferior graft function after transplantation [4,5].

The technology of machine perfusion introduces a new era in organ transplantation. Maintaining an organ under close-to-physiologic conditions not only offers the unique possibility to extend preservation times but also allows for enhanced assessment of organ function [6]. The vision of keeping an organ alive outside the human body for several days opens up completely new perspectives for organ treatment and possibly even organ improvement. While there is evidence that the technology of normothermic machine perfusion (NMP) may promote graft regeneration and reduce IRI by inhibition of inflammation [7], the possibility of adding drugs or cells to the perfusion solution may help to induce organ regeneration [8].

Mesenchymal stem cells (MSCs) may be important tools for this mission. They are multipotent stem cells isolated from different sources such as adipose tissue and bone marrow [9]. MSCs have a wide range of anti-inflammatory and immunomodulatory effects both by cell-to-cell contact and by secreting substances such as chemokines, cytokines, and growth factors [10]. Due to their ability of immunomodulation by suppressing T and B cell proliferation and inhibiting dendritic cell maturation and hence function, MSCs have emerged as a therapeutic cell population in solid organ transplantation [11,12]. Moreover, this specific cell population has been attributed capacities to ameliorate transplantation-related IRI [13]. Based on their multilineage differentiation potential and the capacity to migrate towards damaged tissue, MSCs contribute to tissue repair by secreting bioactive trophic factors, which have a variety of pleiotropic effects, such as enhancing angiogenesis, preventing apoptosis, and fibrosis [14] (Figure 1).

This review summarizes the current state of ex situ machine perfusion and highlights the advantageous combination of this novel technology with the therapeutic potential of MSCs in organ regeneration.

## 2. Bridging Time to Transplantation—From Static Cold Storage to Machine Perfusion

Static cold storage (SCS) remains the established standard for preserving organs prior to transplantation [15]. During organ procurement, the organ is flushed and cooled (0 to 4 °C) with specific preservation fluids, put into plastic bags filled with preservation solution, and stored in an icebox until transplantation. Hypothermia reduces organ metabolism by 10- to 12-fold, and various preservation solutions are available that prevent cell acidosis, cell swelling, and oxygen radical formation; maintain intracellular iron and signal homeostasis; enable electrolyte balance; provide ions and amino acids; contain buffers to maintain physiologic pH and hence preserve cell membrane integrity; and reduce vascular endothelial cell damage [16].

Even though SCS is a well-established, simple, and effective way of organ preservation, it does not allow for organ function testing. While cell metabolism is reduced, the anaerobic metabolism continues at a very low rate, leading to accumulation of succinate and reactive oxygen species (ROS) production upon reperfusion of the graft [17].

Alternative organ preservation methods have been developed in the last decade. Machine perfusion techniques vary with respect to temperature (hypothermic machine perfusion (HMP) at 4 °C to NMP at 37 °C), as well as the composition of the perfusate [18]. HMP operates with a cold preservation solution and continuously perfuses the organ at a controlled, low flow [19]. HMP has been associated with significantly lower rates of delayed graft function (DGF) and an increased 1-year graft survival compared to SCS [20,21,22,23,24].

Over the past years, various NMP devices for the kidney, liver, heart, and lung have been developed for clinical use in transplantation [25,26,27]. A machine used to preserve donor organs under physiological conditions outside the body mimics an in vivo situation and maintains full metabolic organ function. Especially for ECD organs, NMP may be ideal to cautiously bridge time to implantation since it allows for in-depth organ quality assessment [28]. The first in man renal transplantation after NMP was described in 2011 by the group around Hosgood and Nicholson [29]. Shortly after, the first clinical study of NMP in kidney transplantation was published by the same group. Eighteen kidneys from ECD were utilized for NMP prior to transplantation, with a significantly lower rate of DGF in the NMP group, compared to SCS allografts (5.6% vs. 36%) [30]. Weissenbacher et al. demonstrated that urine recirculation during prolonged NMP of discarded kidneys ameliorated metabolic processes and led to enhanced kidney organ function recovery [31].

In 2018, Nasralla et al. performed the first multicenter, randomized controlled trial (RCT) to compare NMP and SCS in liver transplantation. The authors demonstrated that NMP was associated with 50% less graft injury, measured by hepatocellular enzyme release, despite a 50% lower rate of organ discard and a 54% longer mean preservation time [6]. Since then, NMP of the liver has successfully been implemented into the clinical routine and allows for prolongation of preservation times up to 38 h [25].

Primarily in lung transplantation, the discard rate of procured lungs is high due to strict inclusion criteria for acceptable donor lungs. Marginal lung grafts stored with SCS are especially susceptible to IRI, with a high rate of primary graft failure after transplantation [32]. There is evidence that normothermic ex vivo lung perfusion (EVLP) has a beneficial effect on the outcome of these marginal organs. Moreover, transplantation of high-risk donor lungs preserved with EVLP resulted in comparable outcomes to those of conventional (healthy) lung grafts [33,34,35].

As for heart transplantation, the Organ Care System (OCS), which perfuses the donor heart at mild hypothermia (34 °C) with a combination of donor blood and proprietary solution [36], is available for extended heart preservation. The use for ECD hearts resulted in an excellent short-term post-transplant outcome and was comparable to the SCS group [37].

## 3. The Potential of MSCs in Regenerative Medicine

With their capacity to differentiate into multiple cell lineages, MSCs are promising tools in regenerative medicine and tissue engineering [38,39]. Originally, they were characterized from bone marrow (bm) and identified by Friedenstein et al. [40]. Moreover, these multipotent cells can also be found in adipose tissue (at), muscle, peripheral blood, umbilical cord (uc), and placenta, where they support function and repair. The availability of MSCs from multiple sources as well as several advantageous characteristics favor them for clinical use. They are easily accessible and can be expanded simply for large-scale production [41]. Moreover, the capability of self-renewal and differentiation into multiple lineages are important features for tissue regeneration therapies [38,42]. Because of an increasing interest to use MSCs as a potential therapeutic agent in a broad variety of biomedical disciplines, the International Society for Cellular Therapy proposed minimal criteria to define human MSCs. First, MSC must be plastic adherent when maintained in standard culture conditions. Second, they must express CD105, CD73, and CD90 and lack expression of CD45, CD34, CD14 or CD11b, and CD79a or CD19, as well as HLA-DR surface molecules. Third, MSCs must differentiate into osteoblasts, adipocytes, and chondroblasts in vitro (Figure 2) [43].

To exert their therapeutic/regenerative effects, agents or cells must reach their targets. In this context, a major point to consider is the route of administration (systemic delivery (intravenous [i.v.] or intraarterial [i.a.]) or local delivery (e.g., intramuscular [i.m.], intraperitoneal [i.p.], or intracardiac [i.c.])) [39,44,45]. The ideal application route depends on the mechanism of action. Despite some reports indicating that MSCs home to injured tissue independent of the administration route, the majority favor local delivery [46,47,48]. Systemic and direct routes have been investigated in a mouse model. Cell tracking showed prolonged engraftment only after i.m. application [49]. This is in line with a study of Freitas et al. where local injection of bmMSCs and atMSCs in rats with calvarial defects revealed increased bone formation [50].

In theory, the regenerative effect of MSCs is due to their high differentiation potential into cells of ectodermal and endodermal origin (e.g., cardiomyocytes, hepatocytes, or epithelial cells), which then replace damaged and necrotic tissues [51,52,53]. However, there is growing evidence that the impact of MSC-based therapies and their regenerating potential mainly relies on paracrine effects and not on their differentiation into target cells [54,55]. It has been reported that overexpression of either chemokines or their receptors prompts proliferation and migration of administered MSCs and thus enhances the therapeutic effect [56,57,58]. Once MSCs migrate to injured tissue, the secretion of paracrine factors, including chemokines, cytokines, and growth factors, stimulates cells in the near vicinity for tissue repair, thereby exerting antiapoptotic, anti-inflammatory, antifibrotic, and angiogenic effects. These pleiotropic effects of MSCs on injured tissue have been demonstrated in a variety of disease models [59,60,61]. Upon administration of bmMSCs, significantly increased levels of vascular endothelial growth factor (VEGF), fibroblast growth factor (FGF), and insulin-like growth factor (IGF) could be found in injured hearts, resulting in superior regeneration compared to control groups [62]. In an acute kidney injury (AKI) model in rats, the regenerative response of MSCs depended on bioactive factors rather than cell integration and differentiation [63].

The immunomodulatory properties of MSCs are mainly related to the cytokines interleukin-6 (IL-6), IL-10, and macrophage colony-stimulating factor (M-CSF) [64,65]. In a murine sepsis model, bmMSCs induced secretion of anti-inflammatory IL-10, which in turn prevented the differentiation of monocytes into dendritic cells (DC) and hence the full activation of the immune and inflammatory response [66]. The success of MSC-based therapies in experimental settings paved their way for clinical application [12,67]. MSC administration has since been applied in a wide range of diseases such as inflammatory and neurological disorders, diabetes mellitus, ischemic injuries, and graft-versus-host disease (GvHD) [68,69].

## 4. MSC Therapy in Organ Transplantation

The effect of MSCs has been explored in organ transplantation. Therapeutic properties range from anti-inflammatory properties and the potential to mitigate tissue damage to repair and immunomodulation [70,71,72,73]. This implicates the potential for MSC application to reduce IRI [74] and the incidence of acute rejection [12] and promote minimization of immunosuppression or even transplant tolerance [75]. Immunomodulatory properties of MSCs have been studied in a wide range of preclinical small and large animal studies with success in clinical kidney, liver, and lung transplantation [76,77,78] (reviewed in detail elsewhere [79]). Both autologous (bm-derived) and allogeneic (bm-derived and umbilical cord-derived) MSCs have been used clinically. Particularly, allogeneic cells have the potential to modulate an anti-donor immune response and seem more sustainable for clinical implementation since they are readily available [80]. Thus far, MSC studies in transplantation are primarily phase I studies with a focus on safety and feasibility of MSC administration. The first evidence for immunomodulatory effects is indicated by increased regulatory T cells (Tregs) [77,81,82], reduced immune activation towards donor cells [83], and a decrease in immunosuppressive treatment necessary to avoid acute rejection [76]. No toxicity directly related to the MSC infusion, no malignancies, and a comparable number of infectious complications have been reported so far for the use of MSCs in transplant studies [79,84]. Since these trials display a huge heterogeneity in study design, the optimal timing, dosage, and frequency of administration remain to be established [79,85].

## 5. MSCs in Organ Machine Perfusion

Machine perfusion has emerged as a promising technology allowing not only to preserve and comprehensively assess but also to treat and regenerate pre-damaged organs outside the human body. Further to this, NMP may provide a novel way to effectively administer MSCs into the organ (Figure 3). Several studies have shown promising results with respect to the applicability and feasibility of MSC administration during machine perfusion of the kidney, liver, and lung, but little is known about the mechanism through which MSCs may exert beneficial effects. As for the heart, therapeutics such as viral vectors for gene therapy [86] and siRNA [87] have been investigated in preclinical animal trials during ex situ machine perfusion; however, there are no studies available reporting on the use of MSCs during ex situ perfusion of hearts. Table 1 gives an overview of studies investigating safety, feasibility, and efficacy of MSCs in machine perfusion.

### 5.1. MSC Therapy in Machine Perfusion of Kidneys

To test the feasibility of MSC administration during machine perfusion of kidney grafts, the effects of perfusion conditions and fluids on MSCs were assessed by Parraga et al. The group investigated survival, metabolism, and function of thawed cryopreserved human (h)MSCs and porcine (p)MSCs in suspension and demonstrated a 40% reduced pMSC viability in both perfusion fluid and cultured medium, whereas the viability of hMSCs was diminished by only 15% in perfusion fluids. Not surprisingly, the best survival was observed for hMSCs in the cultured medium. The freezing–thawing process impaired viability, thus resulting in reduced adherence to endothelial cells when compared to fresh hMSCs. Thawed hMSCs also showed increased levels of ROS, which may further exert damage on mitochondria. Nevertheless, the potential of MSCs to proliferate and secrete mediators, as well as the secretory profile, was unaffected when cultured with perfusion fluids used for machine perfusion. Most importantly, the therapeutic effect of MSCs was maintained when tested under conditions equivalent to those during NMP [95].

In a next step, porcine kidneys were machine perfused under normothermic conditions for 7 h, and varying numbers of hMSCs (0, 10^5^, 10^6^, 10^7^) were added to the circulating perfusion fluid. Further, fluorescent prelabeled bmMSCs were added to evaluate the localization and viability of MSCs during NMP. The authors demonstrated that the number of circulating MSCs decreased with perfusion time. After 7 h, they were only present in the higher dosage group (10^7^), but they still remained viable. MSCs were localized in the lumen of glomerular capillaries, albeit at a low prevalence. This study demonstrates that the administration of MSCs during NMP is feasible. However, it remains to be elucidated whether the rapid decrease in circulating MSCs is a result of cell migration into the graft, cell death, or adherence within the perfusion system [88].

The group of Lohman et al. demonstrated the feasibility and safety of MSC therapy during NMP prior to transplantation in a porcine autotransplantation model. Porcine kidneys were exposed to warm ischemia and oxygenated HMP, followed by 240 min of NMP, where either 1 × 10^6^ porcine aMSCs or 1 × 10^6^ human aMSCs were added to the perfusion solution. Neither negative effects on perfusion hemodynamics during NMP nor adverse effects on early transplantation outcome were observed. The study provided a proof of concept for ex vivo MSC application during NMP; however, no clear beneficial effect of MSC therapy during the short post-transplant follow-up period could be found [89].

In another porcine kidney NMP model, Pool et al. investigated cytokine levels in the perfusate with and without MSC administration. Porcine kidneys challenged with warm ischemia and subsequent HMP were perfused under normothermic conditions for 7 h. After 1 h of NMP, either 10^7^ human aMSCs or 10^7^ bmMSCs were added. The authors demonstrated that the addition of both types of human MSCs to an ischemically damaged porcine kidney led to an increased release of immunomodulatory cytokines such as human hepatocyte growth factor (HGF), IL-6, and IL-8 into the perfusate, as well as reduced levels of organ injury markers such as lactate dehydrogenase (LDH) and neutrophil gelatinase-associated lipocalin (NGAL), compared to controls without MSC treatment [90].

Brasile et al. used an exsanguinous metabolic support (EMS) platform to investigate the effect of MSCs on IRI of human kidney allografts from DCD donors (mean CIT 29.4 ± 7.4 h) during 24 h of ex situ normothermic perfusion [91]. EMS assembles an acellular medium, a perfusion system, a disposable organ chamber with biosensors to monitor metabolism, and a control module [96,97]. Five human DCD kidney allografts were perfused with EMS for 24 h (control group), and the matching paired kidneys were perfused with EMS, where MSCs were added (1 × 10^8^). A reduced inflammatory response along with an increased synthesis of adenosine triphosphate (ATP) and several growth factors, such as endothelial growth factor, fibroblast growth factor 2, and transforming growth factor α, and normalization of cytoskeleton and mitosis was observed in the MSC-treated group, hinting on tissue regeneration of ischemically damaged human kidney allografts [91].

### 5.2. MSC Therapy in Machine Perfusion of the Liver

There is only a limited number of experimental studies reporting on the use of MSCs in NMP of livers. The group of Rigo et al. established an ex situ NMP model for rats as an experimental setting for organ reconditioning and pharmacological interventions [98]. They were the first to deliver extracellular vesicles from human liver stem cells during a 4-h perfusion period of rat livers and were able to show reduced hepatocellular damage as per histopathology and reduced markers of hepatic cytolysis in the perfusate [99].

Another study reports the use of rat bmMSCs to investigate their beneficial effects on DCD rat livers when administered during ex situ machine perfusion. Compared to the SCS control group, liver function was improved, hepatocyte apoptosis and necrosis were diminished, and mitochondrial damage was reduced in the NMP group with and without bmMSCs. Additional bmMSC administration into the perfusate significantly inhibited intrahepatic macrophage activation and intercellular adhesion, prevented endothelial cell damage, and significantly improved endothelin 1 nitric oxide balance. The authors concluded that the combination of NMP and MSC delivery may improve DCD liver microcirculation and hence organ quality [92]. However, the mechanism of action remains incompletely understood and/or inconclusively demonstrated.

### 5.3. MSC Therapy in Machine Perfusion of the Lung

Ex situ lung perfusion aims to diminish the gap between organ supply and demand and, eventually, to improve the outcome after lung transplantation. Several strategies to improve lung function while the lungs are extracorporeally kept alive on a machine are currently under investigation, one of them being the administration of MSCs during EVLP [34].

In a porcine model of EVLP, Mordant et al. administered human umbilical cord-derived MSCs, either endobronchially or via the pulmonary artery, to determine the favorable route of cell administration. Furthermore, varying dosages of MSCs (5 × 10^7^, 1.5 × 10^8^, or 3 × 10^8^)were tested in order to find an optimal tolerated dose. A significantly higher amount of MSCs remained in the lung parenchyma when administered via the intravascular route, compared to intrabronchial. An optimal dose of 1.5 × 10^8^ was associated with increased concentrations of human vascular endothelial growth factor (VEGF) in lung tissue, as well as decreased IL-8 concentrations in the perfusate [93]. Therapeutic efficacy of MSC application during EVLP was furthermore demonstrated in a study of acute lung injury induced by *E. coli* endotoxin. Human MSCs were added intrapulmonarily (via the trachea) during perfusion to maximize the efficiency of MSC delivery [100], which resulted in an improved alveolar fluid clearance, less lung endothelial permeability, and decreased pulmonary edema [94].

## 6. Considerations, Potential Risks, and Difficulties

Although there is a growing number of experimental studies investigating the administration of MSCs during ex situ machine perfusion in organ transplantation, the actual clinical relevance and the underlying mechanism of their therapeutic effect have not yet been sufficiently understood.

Particularly in solid organ transplantation, there is a major discussion on whether to use recipient- or donor-derived MSCs. Both autologous and allogeneic MSCs appear generally safe and well-tolerated for systemic infusion [101]. While treatment with autologous MSCs has been reported to be safe without provoking unwanted immune responses, allogeneic MSCs are most likely not completely immunoprivileged and may cause cellular and humoral immune responses against donor antigens [102]. In an organ transplant setting, the availability of autologous MSCs still has its limitations. Due to very limited time from the donor organ harvest and allocation until transplantation [103], there is no opportunity to isolate and expand an adequate amount of recipient-derived MSCs in time. Moreover, it is important to culture human MSCs at low clonal density to keep the proliferative capacity [104,105,106]. Development and production of allogeneic MSCs are currently being investigated by pharmaceutical companies, which would allow immediate availability for clinical use [107] and, as a consequence, save time, reduce costs, and promote standardization. The optimal timing of MSC injection may be an important factor to successfully develop cell-based therapy protocols for machine perfused organs. It remains to be elucidated at which time-point MSCs have to be added during the perfusion process. Furthermore, it is not clear whether a single dose of MSCs is sufficient to exert a positive effect or if multiple doses will be favorable. MSC dosages in general and also the timing of application may depend on organ quality, the type of primary disease, and the extent of organ injury/disease. Moreover, the application route within the perfused organ/circuit may vary between different organs (lung vs. liver, vs. heart, vs. kidney).

Besides MSCs, there may also be alternative cell-based therapies to be tested and applied during ex situ machine perfusion in order to ameliorate IRI and to modify, regenerate, or even cure low-quality or diseased organs prior to transplantation. Mesenchymal stem cell-derived extracellular vesicles (MSC-EVs) [108], for example, exert their therapeutic effect by delivering microRNA, mRNA, and various proteins into injured tissue [109]. Moreover, treatments with regulatory T cells (Tregs) [110] and regulatory dendritic cells (DCregs) [111] have emerged as novel therapeutic strategies to ameliorate IRI-related effects and may be considered as potential candidates to be delivered directly into the perfused organ during ex situ perfusion.

Organ treatment with cell therapies may not be completed ex situ due to a limited perfusion period on the machine (several days) and may have to be pursued or repeated at later time-points in the recipient. While extracorporeal treatment on the machine can be highly dosed and even more aggressive, as for tumor or antiviral therapy, further treatment necessary in the recipient may be less aggressive or low-dosed in order to avoid side effects.

Even though progress has been made, optimal selection criteria for MSCs source, immunogenicity, culture conditions, timing, routes of administration, and dosing have yet to be evaluated in the setting of ex situ machine perfusion of organs. Nevertheless, the type of the organ and its dimensions, the extent of organ damage, and the type of disease may also impact the aforementioned criteria of MSC use in machine perfusion.

## 7. Conclusions and Future Perspectives

*Ex situ* machine perfusion introduces a completely new era in the field of organ preservation, treatment, and regeneration. A model of prolonged ex vivo organ preservation may provide a platform to modify, repair, and regenerate organs. Building upon knowledge gained in the field of MSC research, this stem cell type may help to achieve this goal. Until now, only a limited number of studies have reported on the use of MSCs during ex situ machine perfusion, mostly focusing on feasibility and safety aspects. However, no actual clinical benefits have been conclusively demonstrated at this point. Controlled transplantation set-ups are urgently warranted to elucidate any favorable effects and the exact mechanisms of action of MSCs in this setting. As for the future, they may be used not only to treat donor organs in order to increase the donor pool and hence acquire a higher number of ECD organs for transplantation but also to further expand upon curing diseased organs ex vivo.

## Figures and Tables

**Figure 1 ijms-22-05233-f001:**
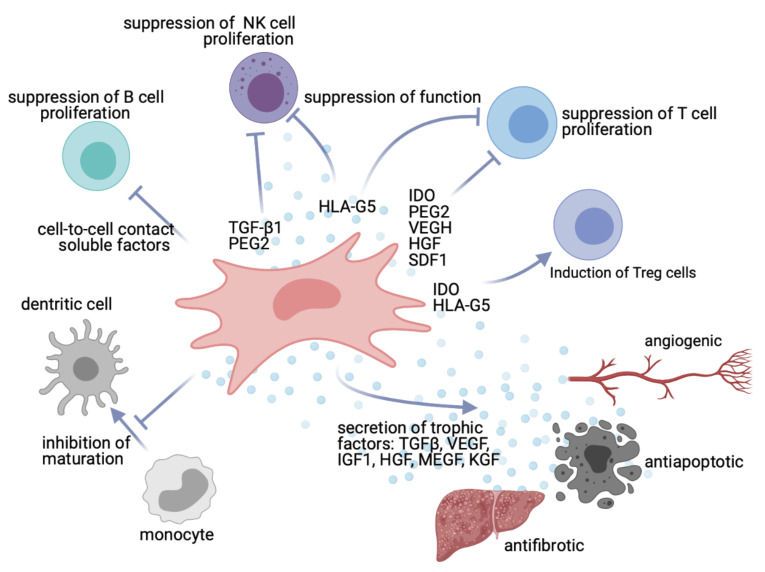
Effects of MSCs on organ regeneration through immunomodulatory and other effects. MSCs suppress T cell, B cell, and natural killer (NK) cell proliferation. They may induce regulatory T cell differentiation via secretion of indoleamine 2,3-dioxygenase (IDO) and human leukocyte antigen-G5 (HLA-G5), have an impact on the maturation of monocytes into dendritic cells, and inhibit NK and T cell function. Through secretion of soluble factors, such as growth factors, cytokines, and chemokines, MSCs contribute to tissue repair, promote angiogenesis, and prevent cell apoptosis and formation of fibrosis. (Figure created with https://biorender.com, accessed on 21 March 2021).

**Figure 2 ijms-22-05233-f002:**
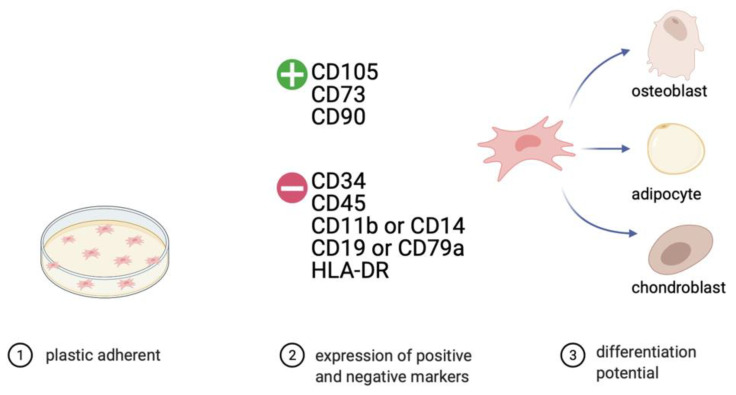
Minimal criteria for defining multipotent mesenchymal stem cells, according to the International Society for Cellular Therapy (ISCT). (Figure created with https://biorender.com, accessed on 21 March 2021). (1) MSCs must be plastic adherent in standard culture conditions. (2) Expression of CD73, CD90, CD105, and absence of the expression of hematopoietic cell surface markers CD34, CD45, CD11b or CD14, CD19 or CD79a, and HLA-DR. (3) Upon specific stimulation, MSCs must differentiate into osteoblasts, adipocytes, or chondrocytes.

**Figure 3 ijms-22-05233-f003:**
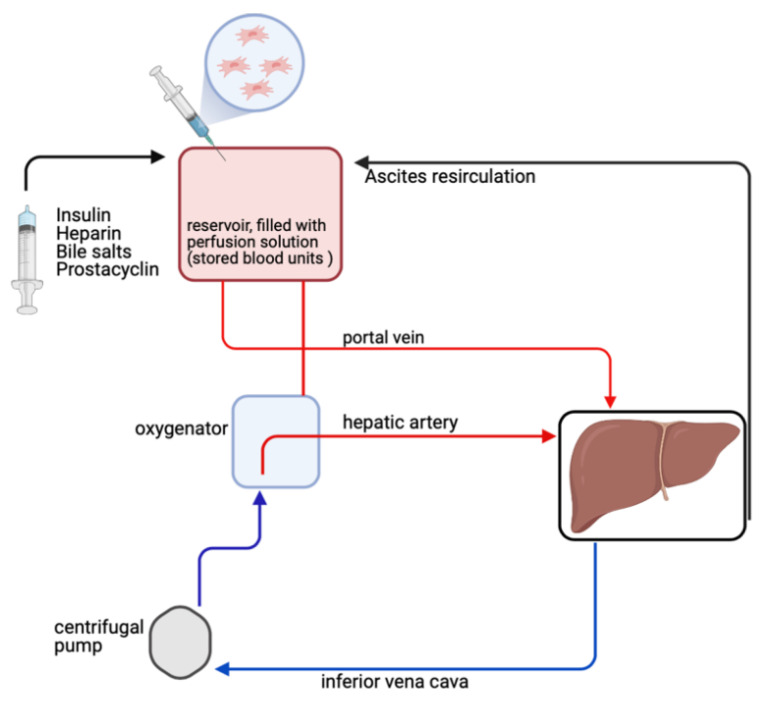
Normothermic machine perfusion as a novel platform to treat and regenerate organs outside the human body. MSCs can be delivered directly into the vasculature of the liver by addition into the circulating perfusate. This concept may help to overcome issues of MSC trafficking and homing (Figure created with https://biorender.com, accessed on 21 March 2021).

**Table 1 ijms-22-05233-t001:** Overview of studies investigating safety, feasibility, and efficacy of MSCs in machine perfusion. To identify articles reporting on MSC therapy during ex situ machine perfusion, a PubMed search using the search terms “mesenchymal stem cell” and “transplantation” and “machine perfusion” was performed on 10 March 2021, thereby identifying 7 studies.

Study	Year	Model	Length of Preservation	Therapeutic Agents	Outcome, Major Findings
Pool et al. [88]	2019	Porcine kidney	7 h of NMP	10^5^ human aMSCs 10^6^ human aMSCs 10^7^ human aMSCs Fluorescent prelabeled bmMSCs	MSCs were detected mainly in the lumen of glomerular capillaries. Minority of glomeruli were positive for fluorescent prelabeled bmMSCs.
Lohmann et al. [89]	2020	Porcine kidney autotransplantation	240 min NMP, after 14 h oxygenated HMP and 75 min WIT	10^6^ porcine aMSCs 10^6^ human aMSCs	Safe and feasible; no beneficial effect could be demonstrated
Pool et al. [90]	2020	Porcine kidney	7 h NMP after 2–3 h of HMP and 20 min WIT	1 × 10^7^ human aMSCs 1 × 10^7^ human bmMSCS	Lower levels of injury markers (Human HGF, NGAL); increased release of immunomodulatory cytokines (IL-6, IL-8, human HGF)
Brasile et al. [91]	2019	Human DCD kidney allografts	24 h of EMS, NMP	10^8^ MSC	Renal regeneration of ischemically damaged kidneys
Yang et al. [92]	2019	Rat liver	NMP	1 × 10^7^ rat bmMSCs	Reduced hepatocyte apoptosis, repaired hepatocyte mitochondrial damage, improvement of histological damage and liver function
Mordant et al. [93]	2016	Porcine lung	EVLP for 12 h after 18 h SCS	5 × 10^7^ ucMSCs endobronchially, or via pulomary artery 1.5 × 10^8^ ucMSCs 3 × 10^8^ ucMSCs	Increased VEGF; decreased IL-8
Lee et al. [94]	2009	Human lung with induced acute lung injury through *E. coli* toxin	EVLP	Human bmMSCs	Increased AFC, decreased endothelial permeability, decreased wet-to-dry ratio

## Data Availability

Not applicable.

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
