# Peer review of "Ex Vivo Mesenchymal Stem Cell Therapy to Regenerate Machine Perfused Organs"

_ijms, 2021, doi:10.3390/ijms22105233_

Round 1

Reviewer 1 Report

This is a very interesting review on a topic that is currently cutting edge.
The manuscript is well written and covers correctly the subject while posing the challenges for the organs concerned.
The heart is missing and this is justified by the type of donors considered.
In the conclusion and perspectives section, can the use of cellular therapy be considered in the recipient to facilitate fonctional recovery or limit side effet of ischemia reperfusion such as fibrosis ou tissue atrophy?

Author Response

Reviewer 1

This is a very interesting review on a topic that is currently cutting edge.
The manuscript is well written and covers correctly the subject while posing the challenges for the organs concerned. The heart is missing, and this is justified by the type of donors considered.
In the conclusion and perspectives section, can the use of cellular therapy be considered in the recipient to facilitate fonctional recovery or limit side effect of ischemia reperfusion such as fibrosis ou tissue atrophy?

Author response 1: We would like to thank the reviewer for the positive feedback and for pointing out some important aspects. After thoroughly screening the literature, there is no data available reporting on the use of MSCs during ex situ perfusion of hearts so far. We have included a statement on this in Chapter 5 (Line 209-212).

We agree with the reviewer that organ treatment with cell therapies may not be completed ex situ during the perfusion process due to a limited perfusion period on the machine (several days), and may have to be pursued or repeated at later time-points in the recipient. While extracorporeal treatment on the machine can be highly dosed or more aggressive, further treatment necessary in the recipient may be less aggressive or low-dosed in order to avoid side effects. We have added a statement on this aspect in Chapter 6 (line 331-334).

Reviewer 2 Report

This paper reviews ex vivo mesenchymal stem cell therapy to regenerate machine perfused organs.

First, possible effects of MSC’s on organ regeneration are presented and the advantages of normal temperature machine perfusion (NMP) compared to static cold storage are highlighted. The second part discusses the potential of MSC’s in regenerative medicine. Finally, the results are given of MSC application in ex vivo machine perfusion of kidney, liver and lung. Apparently no data are given about heart preservation.

Major comment 1: It is unclear how complete the study is: how was the literature screened? It should have been done according to Prisma guidelines.

In theory by local application of MSC’s to organs in NMP their function could be improved.

However, the results are very disappointing sofar:

In kidney some positive effects on surrogate parameters have been obtained (Basile et al, 2019), in liver of rats  some benefit on microcirculation has been reported (Rigo et al, 2018) and in lung only surrogate parameters improved (Mordant et al,2016).

None study shows effect in a controlled transplantation set-up, what will be the proof of the pudding.

Major comment 2: The presented results are rather disappointing and this is not mentioned at the end of the abstract.

Major comment 3: The conclusion of the authors is far too optimistic and not supported by the presented data.

Minor comments

Line 53: secreting

Line 70 : hypothermia

Line 236: histopathology

Line 287: nor (instead of not)

Line 312: M-CSF on the wrong place

Author Response

Reviewer 2

This paper reviews ex vivo mesenchymal stem cell therapy to regenerate machine perfused organs. First, possible effects of MSC’s on organ regeneration are presented and the advantages of normal temperature machine perfusion (NMP) compared to static cold storage are highlighted. The second part discusses the potential of MSC’s in regenerative medicine. Finally, the results are given of MSC application in ex vivo machine perfusion of kidney, liver and lung. Apparently no data are given about heart preservation. 

Author response 1: Thank you for pointing out the missing data about heart preservation. Unfortunately, there are no studies available reporting on the use of MSCs cells during ex situ perfusion of hearts so far. We have addressed this in Chapter 5 (Line 209-212).

Major comment 1: It is unclear how complete the study is: how was the literature screened? It should have been done according to Prisma guidelines. In theory by local application of MSC’s to organs in NMP their function could be improved.

However, the results are very disappointing sofar: In kidney some positive effects on surrogate parameters have been obtained (Basile et al, 2019), in liver of rats some benefit on microcirculation has been reported (Rigo et al, 2018) and in lung only surrogate parameters improved (Mordant et al,2016). 

None study shows effect in a controlled transplantation set-up, what will be the proof of the pudding.

Author response to major comment 1:  Thank you for the critical evaluation of our manuscript.

We agree with the reviewer´s comment that the Prisma guidelines should be applied when conducting a systematic review or meta-analysis. However, as only few studies investigating the use of MSCs in machine perfusion are available so far, which significantly vary in study design and readout, a more conclusive systemic review or a meta-analysis is precluded. However, in order to better document the way how the literature has been screened, relevant information has been included in the legend to Table 1 (Line 293-295).

We further agree with the reviewer that clear demonstration of the beneficial effects of MSC therapy are missing at this point. However, this is mainly attributed to the very early stage of research conducted in this field. As summarized in this review, studies mainly focused on dosage finding, feasibility and safety aspects. We also agree that coordinated studies with a controlled set-up and readout have to be conducted in order to further develop the idea of regenerating or even curing diseased organs with the help of MSCs while machine perfusion. We have stated this in Chapter 7 (Conclusions) (Line 345-349).

Major comment 2: The presented results are rather disappointing, and this is not mentioned at the end of the abstract.  

Author response to major comment 2:  We would like to thank the reviewer for the comment. To meet the reviewers concerns we have added a paragraph at the end of the abstract mentioning the limitations of the studies presented here and that now clear beneficial effect of MSCs in machine perfused organs has been observed so far (Line 22-26).

Major comment 3: The conclusion of the authors is far too optimistic and not supported by the presented data.

Author response to manor comment 3: Thank you for your valuable comment. We have modified the conclusion and hope that the reviewer finds it appropriate.

Minor comments

Line 53: secreting

Line 70: hypothermia

Line 236: histopathology

Line 287: nor (instead of not)

Line 312: M-CSF on the wrong place

Author response to minor comments: Thank you for your valuable input. The above-mentioned typos have been corrected.

Reviewer 3 Report

Dear authors,

The authors wrote a review about the use of MSC to regenerate machine perfused organs. It's an approach that is developed for the past few years, to extend the storage of the organs, to overcome the shortage of organ donors.

I have few comments:

1) All latin words should be in italic (via, etc...)

2) Typo to be verified (e.g line 170 space, line 255 Theurapeutic...) and the font is not homogenous over the manuscript.

3) In general, MSC are used. Could the authors mention which MSC are used in the example mentioned in the manuscript?

4) Usually, MSC are injected after  the perfusion, and the dose and the route of injection is important. but the timing of the injection also. Could the authors comment on the timing?

5) the authors mentioned that it can be very difficult to harvest MSC for autologous injection, and this might require the development of allogeneic MSC production for injection. Another point, the authors should develop (mentioned shortly in the manuscript) is the use of extracellular vesicles, produced by MSC.

Author Response

Reviewer 3

The authors wrote a review about the use of MSC to regenerate machine perfused organs. It's an approach that is developed for the past few years, to extend the storage of the organs, to overcome the shortage of organ donors. I have few comments:

Reviewer comment 1: All latin words should be in italic (via, etc...)

Author response 1:  We thank the reviewer for pointing this out. All the Latin words have been changed to italic.

Reviewer comment 2: Typo to be verified (e.g line 170 space , line 255 Theurapeutic...) and the font is not homogenous over the manuscript.

Author response 2:  We thank the reviewer for the comment. Typos have been corrected and the font has been changed to appear homogenously over the manuscript.

Reviewer comment 3: In general, MSC are used. Could the authors mention which MSC are used in the example mentioned in the manuscript?

Author response 3:  We would like to thank the reviewer for pointing this out. We have added the missing information on which type of MSCs (species specific) or where they were derived from (bone marrow, umbilical cord, adipose tissue) in the provided table 1. Unfortunately, one study (Brasile et al.) did not mention the exact origin of MSCs they used.

Reviewer comment 4: Usually, MSC are injected after the perfusion, and the dose and the route of injection is important. but the timing of the injection also. Could the authors comment on the timing?

Author response 4:  Thank you for pointing this out. We agree with the reviewer that the timing of injection may be an important factor when developing such cell-based therapy protocols. Despite promising results of some studies, there is limited data on the optimal timing of MSCs injection during machine perfusion of organs. Timing, dosage and application route may vary also between different organs (kidney, vs lung, vs liver vs heart) and remains to be under investigation. However, we have addressed this very important aspect, and have added a section on the timing in Chapter 6 (Line 311-313).

Reviewer comment 5: the authors mentioned that it can be very difficult to harvest MSC for autologous injection, and this might require the development of allogeneic MSC production for injection. Another point, the authors should develop (mentioned shortly in the manuscript) is the use of extracellular vesicles, produced by MSC.

Author response 5:  Thank you for raising this very interesting and important aspect. The possibility to use extracellular vesicles, produced by MSCs, as well as alternative cellular based therapies to be administered in the course of ex situ organ perfusion, has been implemented in Chapter 6 (Line 324-330).

Round 2

Reviewer 2 Report

Suggestions have been followed. The conclusion is now appropriate.

There is a new typo: line 319 machanisms  instead of mechanisms

Reviewer 3 Report

Dear Authors,

After reading the authors responses, I have no additional comments. I accept the manuscript as it is.

Sincerely.